# Organic Fertilizer Sources Distinctively Modulate Productivity, Quality, Mineral Composition, and Soil Enzyme Activity of Greenhouse Lettuce Grown in Degraded Soil

Mariateresa Cardarelli [1] , Antonio El Chami [1], Paola Iovieno [2], Youssef Rouphael [3] , Paolo Bonini [4]
and Giuseppe Colla [1,*]

1 Department of Agriculture and Forest Sciences, University of Tuscia, 01100 Viterbo, Italy
2 CREA—Consiglio per la Ricerca in Agricoltura e l'Analisi dell'Economia Agraria, Centro di Ricerca Orticoltura e Florovivaismo, 84098 Pontecagnano Faiano, Italy
3 Department of Agricultural Sciences, University of Naples Federico II, 80055 Portici, Italy
4 oloBion S.L., 08028 Barcelona, Spain
* Correspondence: giucolla@unitus.it

**Abstract:** Intensive greenhouse vegetable production is often associated with a decline of crop productivity due to the increase of soil salinity and/or a reduction of biological fertility. The aim of the current work was to assess the effects of three organic fertilizers on morpho-physiological and agronomic traits of greenhouse lettuce as well as soil enzyme activity under poor soil quality conditions. The tested organic fertilizers (poultry manure, vinasse-based fertilizer, and insect's frass fertilizer) were applied pre-planting at the same equivalent nitrogen (N) rate (90 kg N ha$^{-1}$). Laboratory incubation assay results showed that vinasse-based fertilizer was the most suitable fertilizer in supplying the mineral N in the short term. All fertilizers increased shoot fresh and dry weight compared to unfertilized control with a more pronounced effect (+75%) with vinasse-based fertilizer and insect's frass. Insect frass reduced by 27% the leaf nitrate concentration in comparison with the other treatments. The toxic heavy metal Pb was 46% lower in all organically fertilized lettuce leaves. Soil enzymatic activities of acid phosphatase, alkaline phosphatase, arylsulfatase (ArS), N-acetyl-β-D-glucosaminidase (NAGase), dehydrogenase, and total hydrolase (THA) were enhanced by poultry manure and insect's frass in comparison with unfertilized control while vinasse-based fertilizer increased ArS, NAGase, and THA. Taken together, our data demonstrate that the application of organic fertilizers especially vinasse-based fertilizer and insect's frass during intensive crop production is a suitable approach for mitigating the negative impact of soil salinity, enhancing soil biological fertility, and improving agronomic performance of greenhouse lettuce.

**Keywords:** organic fertilizers; *Lactuca sativa* L.; physiological traits; produce quality; enzyme activity; sustainable horticulture

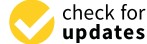



## 1. Introduction

Finding efficient means to optimize crop growth and quality without squandering priceless natural resources is one of the major issues affecting the entire food supply chain today. The preservation of significant non-renewable resources, such as soil, should come after efforts to reduce chemical (e.g., synthetic) inputs and maintain production quality. Soil quality and fertility are being negatively impacted by anthropogenic activity and climate change, which also result in issues with organic matter loss, rising salinity from chemical abuse, and drought from insufficient precipitation [1]. Traditional and intensive cultivation methods can increase the salt content in the soil, and crops cultivated directly in the soil can be negatively affected by low content of organic matter. Soil salinization and loss of organic matter are the main causes of soil degradation in vegetable crop production. The public widely criticized the greenhouse industry for using unnatural production techniques and for severely polluting

the environment with the residues of fertilizers, pesticides, and other agrochemicals used [2]. Therefore, a sustainable intensification of crop production which aims maintaining a good soil fertility is very important for assuring long-term crop productivity.

The European Union is making serious efforts to provide tools to reduce agriculture's environmental impact, starting with legislation [3]. The circular economy foundations encourage the reuse and valorization of by-products from the agricultural and livestock industry to create suitable products for cultivations, such as organic fertilizers. The rise of organic agriculture is one of the main efforts to make this sector more sustainable, increasingly avoiding synthetic fertilizers and pesticides [4]. Several studies have shown that using organic fertilizers results in an improvement of soil physical properties, soil enzyme activity, soil organic carbon, and an increase in nutrients which leads to enhance microbial development, thus increasing soil fertility and quality [5–8]. Soil enzymes and soil microorganisms play an active role in influencing soil fertility due to their participation in the cycle of nutrients, such as carbon and nitrogen which are required for plant growth, but they are also sensitive biological indicators for soil quality evaluation in addition to sensitively reflecting minute changes in soil environment [9,10].

Sustainable organic fertilization relies on understanding how the quality of organic fertilizer affects the synchrony of nitrogen (N) supply and crop N demand which is critical for optimal agronomic N management, environmental protection, and crop productivity, especially in short crop cycles [11,12]. Some authors investigated the rate of soil respiration and the dynamics of the mineral N in various soil/organic fertilizer mixtures in a laboratory incubation experiment [13]. When compared to urea, organic fertilizers significantly increased soil respiration and slowed the release of mineral N in the soil. Moreover, the dynamic of mineral N released from organic fertilizers varied significantly during the incubation period (12 weeks) with a fast mineral N release at 25 °C in the organic fertilizers containing feather meal, vinasse, guano, and poultry manure. Organic fertilizers with a fast release of N are necessary to meet the N requirements of short life cycle crops such as lettuce. Lettuce is a fast-growing crop which requires a continuous and sufficient supply of nutrients during all growing cycles. Moreover, lettuce is sensitive to soil salinity [14] causing growth depression and physiological disorder known by lettuce tip burn that is due to a problem of calcium uptake and transportation in leaves during periods of rapid growth [15]. In addition, as a leafy crop, lettuce is very susceptible to nitrate accumulation in its leaves due to the oversupply with nitrogen fertilizers [16,17]. Nitrate accumulation in edible plant tissue has a significant impact on human health; when it is converted to nitrite in the body, it can result in respiratory dysfunction called methemoglobinemia, which is harmful to children [18]. Furthermore, it might also oversee the formation of nitrosamines, which are known to cause cancer [19].

Researchers have focused on investigating the effects of different organic fertilizers on lettuce growth, yield, leaf nitrate content, and soil fertility, while limited studies have considered the agronomic effect of organic fertilizers under low fertile soils resulting from intensive greenhouse horticulture. Moreover, limited studies have considered newly insect frass fertilizers derived from the digestate of black soldier fly larvae. A group of authors evaluated the potential agronomic value of black soldier fly larvae frass (BSFF) as organic fertilizer in greenhouse lettuce production under pot conditions in comparison with mineral fertilizers at different doses. The BSFF increased the soil's organic matter and residual nutrient content after 42 days of the experiment, as well as the enzymatic activity of dehydrogenase and β-glucosidase, while slow mineralization rate of N afforded by BSFF resulted in the lowest yield compared to exclusive mineral fertilization and a mixture of organic/mineral fertilization [20].

On the contrary, other studies reported an increase of lettuce yield with application of several organic fertilizers from plant by-products, biochar, and olive waste derived compost. These findings highlighted the agronomic benefits of organic fertilizers if mineral N released from organic fertilizers is synchronized with the nutritional demands of lettuce crop [21,22]. Lettuce quality traits such as leaf area, and mineral, chlorophyll, and nitrate contents were

also improved with application of several organic fertilizers such as biodigested vinasse, digested livestock manure, and liquid organic fertilizer obtained from powdered soybeans, rice bran, powdered seaweed, molasses, and commercial microorganisms [23–26]. Additional studies demonstrated that soil application of organic materials like food waste compost, garlic stalk by-product, and poultry litter biochar enhanced soil enzyme activity in lettuce cultivation [22,27–31]. However, the above studies have not considered a comparison between the most promising fast-acting organic fertilizers such as poultry manure, vinasse-based fertilizer, and insect frass in greenhouse lettuce production under reduced soil fertility stemming from intensive management.

The aim of this work was: (1) to explore the dynamic of mineral N release in soil amended with three organic fertilizers such as poultry manure and vinasse-based fertilizer with the newly developed insect's frass fertilizer under controlled environment (Exp. 1—Laboratory soil incubation assay); (2) to evaluate the effects of the three tested organic fertilizers on morpho-physiological and agronomic traits of greenhouse lettuce crop and soil enzyme activity under low soil quality conditions (Exp. 2—Greenhouse trial).

## 2. Materials and Methods

### 2.1. Laboratory Soil Incubation Study (Exp. 1)

#### 2.1.1. Experimental Conditions

An incubation test was carried out at the Horticulture Biotechnology Lab of University of Tuscia to quantify the mineral N release rate of the three organic fertilizers. Soil samples were collected at a depth of about 15 cm in the greenhouse used for the agronomic trial on lettuce for determining the N mineralization. After air-drying, greenhouse soil was screened through a 5 mm sieve. The three tested organic fertilizers in the form of dried pellets were: poultry manure (40.0 g N/kg, 17.5 g P/kg, 33.2 g K/kg; organic matter content 707 g/kg; C:N ratio 10.2), insect's frass (30.0 g N/kg, 13.1 g P/kg, 24.9 g K/kg; organic matter content 750 g/kg; C:N ratio 14.3), and a vinasse-based fertilizer containing mainly dried vinasse with the addition of poultry manure, guano and feather meal (60.0 g N/kg, 34.9 g P/kg, 124.5 g K/kg; organic matter content 500 g/kg; C:N ratio 4.8). Poultry manure (Italpollina), insect's frass (Laphrassea), and vinasse-based fertilizer (Phenix) were provided by Hello Nature Company (Rivoli Veronese, Italy). Insect's frass was derived from the digestate of black soldier fly larvae. Fertilizers were ground with a mill. Soil samples of 40 g each were prepared and mixed with each ground fertilizer at the following rate 10.0, 7.5, and 5.0 mg/g of soil for insect's frass, poultry manure, and vinasse-based fertilizer, respectively; these fertilizer rates provided the same amount of N in all tested soil:fertilizer mixtures. An unfertilized control was also included in the trial. Soil and soil:fertilizer mixtures were placed in jars (diameter 6 cm$^2$) and watered to 70% of water container capacity.

#### 2.1.2. Experimental Design and Analysis

The four treatments (three soil:fertilizer mixtures with insect's frass, poultry manure, and vinasse-based fertilizer, and unfertilized control) were compared in a randomized complete design with eight replicates (jars). Jars were incubated for 2 or 5 weeks in a cabinet thermostated at 24 °C in dark conditions, covering the jars with punched aluminum foil to aerate the samples. A total of 64 jars (4 treatments × 8 replicates × 2 incubation times) were included.

After 2 and 5 weeks of incubation, 8 moist samples of the soil blends for each treatment were diluted with 0.01 M CaSO$_4$ and stirred for 60 min before being centrifuged at 3000× $g$ rpm for 5 min. The surnatant was used to determine mineral N in form of nitrate (N-NO$_3$) [32] and ammonium (N-NH$_4$) [33].

Percentage of N mineralization from the amendments was calculated as the percentage of total mineral N minus the quantity from jars of nonamended soil blends divided by the total N provided by the amendment.

*2.2. Greenhouse Trial (Exp. 2)*

2.2.1. Experimental Conditions

The trial was carried out in a 200 m$^2$ polyethylene greenhouse at the Experimental Farm of Tuscia University (42°25′ N, 12°08′ E, 310 m a.s.l.) in autumn 2019. The soil was previously cultivated with vegetable crops such as melon, tomato, cucumber, and chicory. The soil contained 64% sand, 20% silt, and 16% clay. Soil chemical characteristics were: pH 7.3, electrical conductivity of saturated paste extract 5.47 dS/m, organic matter 18.8 g/kg, total nitrogen 1.16 g/kg, cation exchange capacity 25.3 meq/100g; exchangeable cations was as follow (mg/kg): K (804.1), P (15.3), Ca (3420), Mg (420), Na (502). Soil trace elements were as follow (mg/kg): soluble B (1.42), assimilable Fe (5.0), assimilable Mn (20), assimilable Cu (16.6), assimilable Zn (4.8). Based on the physico-chemical analysis, the soil was classified as sandy loam, with neutral pH, high cation exchange capacity, high salinity, and low organic matter content; moreover, the soil was very high in exchangeable K, Ca, and Mg, and available P, high in exchangeable Na, soluble B, and available Cu, medium in total N, available Mn and Zn, and low in available Fe. The above findings are in line with greenhouse soils resulting from intensive vegetable production where the intensive tillage and high temperatures lead to fast mineralization of organic matter while the lack of rainfall together with the overuse of mineral fertilizers lead to soluble salt build-up in the topsoil. The butterhead lettuce (*Lactuca sativa* L.) cultivar "Fabietto" (Rijk Zwaan, Bologna, Italy) was chosen as test crop for its early development and high productivity. The seedlings were transplanted at two-true leaf stage on 2 October in single rows (0.3 × 0.3 m) with a plant density of 11 plants m$^{-2}$. Plants were grown under natural light conditions, and the daily temperatures were maintained between 12 and 25 °C. The irrigation was performed with a drip irrigation system with in-line emitters located 0.30 m lines apart and an emitter flow rate of 3.4 L h$^{-1}$. Drip lines were placed in the middle of each inter-row.

2.2.2. Experimental Design, Measurements, and Analysis

The three organic fertilizers previously described [poultry manure (Italpollina), insect's frass (Laphrassea), and vinasse-based fertilizer (Phenix)] in the Exp. 1 were compared in a randomized complete block design with 4 replicates. An unfertilized control was also included. Each plot contained 4 rows of 10 plants each; the trial included a total number of 16 plots.

Organic fertilizers in the form of dried pellets were buried into the surface soil layer (0–15 cm) one day before transplant at the following rates: 2.25 t ha$^{-1}$ of poultry manure, 3.0 t ha$^{-1}$ of insect frass, and 1.5 t ha$^{-1}$ of vinasse-based fertiliser; these organic fertilizer rates provided the same nitrogen (N) equivalent dose of 90 kg N/ha. The organic matter (OM) and total carbon (TC) supplied by the three organic fertilizer rates were as follow (t ha$^{-1}$): poultry manure (1.93 OM; 0.92 TC), insect frass (2.64 OM; 1.29 TC), vinasse-based fertilizer (0.92 OM; 0.43 TC).

During the growing cycle, the Soil Plant Analysis Development (SPAD) index was measured on leaves as indicator of the chlorophyll content with the SPAD-502 instrument (Konica Minolta Europe). The readings were taken 4, 9, 15, 23, 28 days after transplanting on 20 topmost fully expanded leaves per plot. Chlorophyll fluorescence of leaves was measured with Handy-PEA (Hansatech Instruments Ltd., King's Lynn, Norfolk, UK) to determine the maximum quantum efficiency of photosystem II (Fv/Fm). The readings were taken on 20, 25, and 30 days after transplanting on 3 topmost fully expanded leaves per plot.

Lettuce plants were harvested at commercial maturity stage, after 31 days of transplanting. The yield was measured as shoot fresh weight. Shoot dry weight, root dry weight, and root to shoot ratio were determined after oven-drying plant tissues at 65 °C until constant sample weight.

Dry leaf samples harvested at the end of the trials were ground separately in a Wiley mill to pass through a 20-mesh screen, then 0.5 g of the dried plant tissues were analyzed for the following mineral elements: N, P, K, Ca, Mg, Fe, Mn, Zn, Cu, Ni, Na, Al, Ba, Cd, Co, Cr, Pb, Se, and Sn. Nitrogen concentration in the plant tissues was determined after mineralization with sulfuric acid by 'Kjeldahl method' [34] while the other elements were

determined by dry ashing at 400 °C for 24 h, dissolving the ash in 1:20 $HNO_3$, and assaying the solution obtained using an inductively coupled plasma emission spectrophotometer (ICP Iris; Thermo Optek, Milano, Italy) [35].

At the end of the experiment, soil samples were randomly collected in the topsoil (0–0.15 m) within each plot by using a metal soil probe (2.5 cm diameter). Ten cores were collected from each plot and combined into one composite sample and homogenized through a 2 mm sieve. Soil samples were refrigerated at 4 °C and transported to laboratory to determine dry weight, active carbon, and soil enzyme activity. Each analysis was carried out on two laboratory replicates.

Five grams of fresh soil was weighed, and oven dried at 105 °C until constant weight to determine soil dry weight. Soil samples were air-dried and further sieved (1 mm) before labile soil carbon determination [36]. According to the method, the $KMnO_4$ stock solution (0.2 M) was diluted 1:10 with distilled water and mixed to soil samples (20 mL of solution and 5 g of dried soil). The soil $KMnO_4$ mixture was agitated for 2 min at 120 rpm and then allowed to stand for 10 min. Once the soil was well deposited, 0.50 mL of surnatant was added to 45 mL distilled water and shaken 10 s. Each diluted solution was then poured into an optically calibrated glass vial and the light absorption was measured at a 550 nm fixed wavelength using a UV-Vis Shimadzu, UV-1800 spectrophotometer (Shimadzu Corporation, Kyoto, Japan).

Acid phosphatase (AcP, EC 3.1.3.2) and alkaline phosphatase (AlkP, EC 3.1.3.1) activities were assayed by the hydrolysis rate of *p*-nitrophenyl disodium orthophosphate (*p*-NPP) supplied as substrate [37,38]. Arylsulphatase (ArS, EC 3.1.6.1) was assayed using *p*-nitrophenyl sulphate (*p*-NPS) as substrate. Dehydrogenase (DHA, EC 1.1.1) activity was assayed using tetrazolium salt as substrate [39] and N-acetyl-β-D-glucosaminidase (NAGase, EC 3.2.1.30; EC 3.2.1.52) activity was determined using *p*-nitrophenyl-N-acetyl-β-D-glucosaminidine as substrate [40]. In addition, total hydrolytic activity (THA) was assessed by the fluorescein diacetate method that measures activities of several enzymes such as proteases, lipases and non-specific esterases, related to organic matter decomposition [41]. All spectrophotometric measurements were carried out by UV-1800 Shimadzu spectrophotometer. The enzyme activities were expressed as μg per gram of dry soil per hour.

### 2.3. Statistical Analysis

All data were subjected to ANOVA using the SPSS22 software package (Chicago, IL, USA). Means were separated using Tukey's range test performed at a 5% level of significance. For each enzyme, Person correlation was performed between enzyme activity and organic fertilizer rate. A hierarchical cluster analysis on the morpho-physiological crop traits, soil enzymatic activity and soil active carbon was performed, and a heatmap was generated using the ClustVis online tool. Matrix values were normalized as ln(x + 1), with Euclidean distance and complete linkage.

### 3. Results

### 3.1. Nitrogen Mineralized from Organic Fertilizers during Laboratory Incubation Assay

To determine the nitrogen release pattern from poultry manure, insect frass and vinasse-based fertilizers, a laboratory aerobic incubation experiment was conducted for 5 weeks, and the outcomes are shown in Figure 1. After 2 weeks of incubation, vinasse-based fertilizer had highest N mineralization followed by poultry manure and insect frass. After 5 weeks of incubation, only vinasse-based fertilizer had higher significant N mineralization compared to insect frass while poultry manure has an intermediate value.

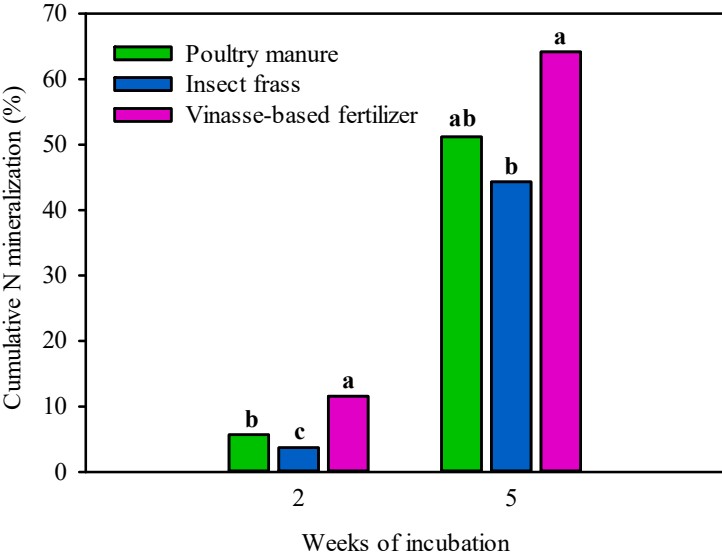

**Figure 1.** Mean cumulative N mineralization by fertilizer type at 2 and 5 weeks of aerobic incubation of soil: fertilizer blends. Mean square error at 2 wk = 0.53; Mean square error at 5 wk = 145.85; Coefficient of variation at 2 wk = 46.8%; Coefficient of variation at 5 wk = 26.9%; Tukey's least significant difference (5%) at 2 wk = 0.92; Tukey's least significant difference (5%) at 5 wk = 15.22 Different letters within each week of incubation indicate significant differences according to Tukey's range test (5%).

### 3.2. Agronomic Response of Lettuce and Soil Enzymatic Activity under Greenhouse Trial

Organic fertilizers significantly ($p < 0.001$) affected chlorophyll concentration, expressed as SPAD index, of lettuce leaves at 9 and 15 days after transplanting (DAT) while no significant differences were found among treatments at 4 DAT (Table 1). At 9 and 15 DAT, the highest SPAD index was recorded in plant leaves treated with vinasse-based fertilizer followed by insect frass and poultry manure which showed similar values; the lowest SPAD value was recorded in untreated control.

**Table 1.** Effect of fertilizer treatments on chlorophyll concentration expressed as SPAD index of lettuce leaves.

| Fertilizer Treatment | SPAD Index | | | | |
|---|---|---|---|---|---|
| | **4 DAT** | **9 DAT** | **15 DAT** | **23 DAT** | **28 DAT** |
| Control | 14.93 | 19.52 c | 24.67 c | 24.90 | 24.65 |
| Poultry manure | 14.53 | 23.30 b | 26.27 b | 24.31 | 24.67 |
| Insect frass | 15.46 | 23.13 b | 26.33 b | 24.38 | 24.88 |
| Vinasse-based fertilizer | 15.25 | 24.63 a | 28.29 a | 25.20 | 25.24 |
| MSE | 0.35 | 0.39 | 0.41 | 0.64 | 1.13 |
| CV (%) | 4.27 | 8.99 | 5.47 | 3.28 | 3.26 |
| Tukey (5%) | 1.25 | 1.32 | 1.34 | 1.68 | 2.24 |

MSE = mean square error; CV = coefficient of variation; Tukey (5%) = Tukey's least significant difference (5%). Different letters within each column indicate significant differences according to Tukey's range test (5%).

The outcomes of the effect of organic fertilizers on maximum quantum yield of PSII (Fv/Fm) of lettuce leaves revealed that the application of the three organic fertilizers had no effect on Fv/Fm at 20 and 30 DAT while a significant ($p < 0.05$) decrease of Fv/Fm value was recorded at 25 DAT with poultry manure application in comparison to other organic treatments and unfertilized control (Table 2).

**Table 2.** Effect of fertilizer treatments on maximum quantum yield of PSII (Fv/Fm) of lettuce leaves.

| Fertilizer Treatment | Fv/Fm | | |
|---|---|---|---|
| | **20 DAT** | **25 DAT** | **30 DAT** |
| Control | 0.837 | 0.831 a | 0.852 |
| Poultry manure | 0.836 | 0.822 b | 0.845 |
| Insect frass | 0.842 | 0.834 a | 0.855 |
| Vinasse-based fertilizer | 0.838 | 0.830 ab | 0.849 |
| MSE | 0.000279 | 0.000014 | 0.000465 |
| CV (%) | 1.804 | 0.577 | 2.313 |
| Tukey (5%) | 0.035 | 0.008 | 0.045 |

MSE = mean square error; CV = coefficient of variation; Tukey (5%) = Tukey's least significant difference (5%). Different letters within each column indicate significant differences according to Tukey's range test (5%).

The results of the effect of fertilizer treatments on morpho-physiological and agronomic traits of lettuce crop are shown in Table 3. The application of the three organic fertilizers affected significantly both the shoot fresh ($p < 0.001$) and dry weight ($p < 0.01$) while no significant differences were found on the root dry weight and the root to shoot ratio (Table 3). The highest value of shoot fresh weight was obtained in plants treated with vinasse-based fertilizer and insect frass in comparison with the control while the poultry manure treatment gave intermediate values.

**Table 3.** Effect of fertilizer treatment on shoot fresh weight, shoot and root dry weight, and root to shoot ratio of lettuce plants.

| Fertilizer Treatment | Shoot Fresh Weight (g Plant$^{-1}$) | Shoot Dry Weight (g Plant$^{-1}$) | Root Dry Weight (g Plant$^{-1}$) | Root to Shoot Ratio |
|---|---|---|---|---|
| Control | 132.31 c | 5.13 b | 1.91 | 0.39 |
| Poultry manure | 172.99 b | 7.49 ab | 2.46 | 0.34 |
| Insect frass | 230.99 a | 9.97 a | 2.41 | 0.24 |
| Vinasse-based fertilizer | 233.15 a | 9.33 a | 2.70 | 0.29 |
| MSE | 331.5 | 2.01 | 0.22 | 0.01 |
| CV (%) | 24.21 | 29.05 | 21.80 | 31.42 |
| Tukey (5%) | 38.24 | 2.98 | 1.00 | 0.19 |

MSE = mean square error; CV = coefficient of variation; Tukey (5%) = Tukey's least significant difference (5%). Different letters within each column indicate significant differences according to Tukey's range test (5%).

Among the tested organic fertilizers, insect's frass significantly ($p < 0.05$) reduced the nitrate concentration in lettuce leaves in comparison with the other organic fertilizers while untreated control gave intermediate values which were not significantly different from the organic fertilizer treatments (Figure 2).

Except for the leaf P concentration which was significantly ($p < 0.01$) highest in vinasse-based fertilizer treated plants compared to the other fertilizer treatments and control, the soil amendment with the three organic fertilizers did not affect the macronutrient concentrations in lettuce leaves (Table 4).

Apart from for Mo, which was significantly ($p < 0.01$) lowest in vinasse-based treatment, no significant differences were recorded for the other micronutrients in lettuce leaves (Table 5).

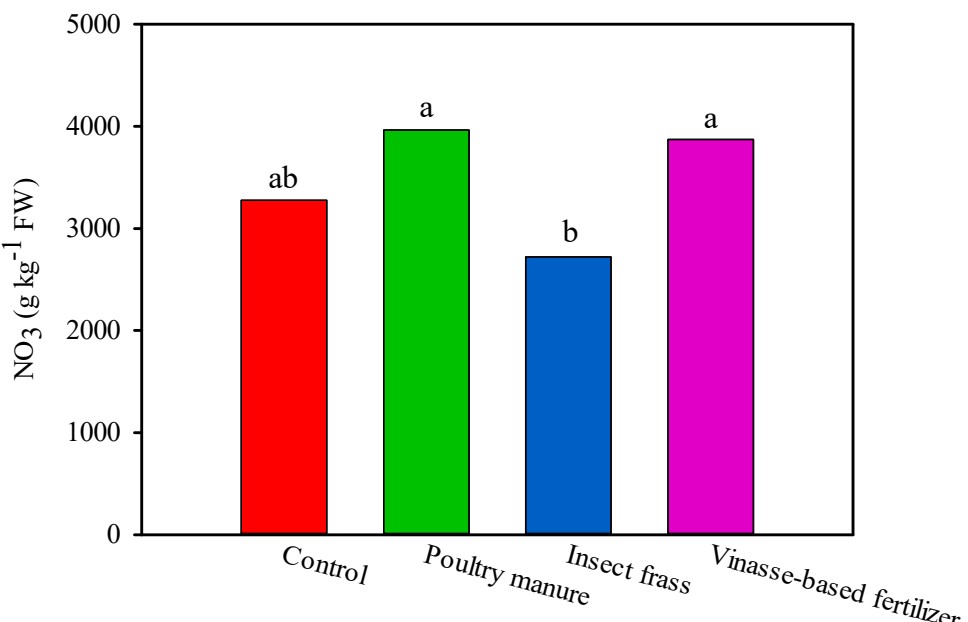

**Figure 2.** Effect of fertilizer treatments on nitrate concentration in lettuce leaves. Mean square error = 292,554; Coefficient of variation = 20.5%; Tukey's least significant difference (5%) = 1135.9. Different letters indicate significant differences according to Tukey's range test (5%).

**Table 4.** Effect of fertilizer treatments on macronutrient concentration in lettuce leaves.

| Fertilizer Treatment | Macronutrients (g/kg DW) | | | | |
|---|---|---|---|---|---|
| | N | P | K | Ca | Mg |
| Control | 44.16 | 3.33 b | 82.72 | 14.09 | 5.63 |
| Poultry manure | 43.10 | 4.05 b | 90.30 | 12.93 | 5.95 |
| Insect frass | 41.05 | 4.07 b | 84.36 | 12.51 | 5.78 |
| Vinasse-based fertilizer | 42.55 | 5.09 a | 85.52 | 11.80 | 5.51 |
| MSE | 3.09 | 0,20 | 40.67 | 1.78 | 1.29 |
| CV (%) | 4.57 | 18.51 | 7.47 | 11.45 | 18.02 |
| Tukey (5%) | 3.69 | 0.95 | 13.39 | 2.80 | 2.39 |

MSE = mean square error; CV = coefficient of variation; Tukey (5%) = Tukey's least significant difference (5%). Different letters within each column indicate significant differences according to Tukey's range test (5%).

**Table 5.** Effect of fertilizer treatments on essential trace elements in lettuce leaves.

| Fertilizer Treatment | Essential Trace Elements (mg/kg DW) | | | | | | |
|---|---|---|---|---|---|---|---|
| | B | Fe | Mn | Zn | Cu | Mo | Ni |
| Control | 21.72 | 190.89 | 285.36 | 52.40 | 12.78 | 0.37 a | 0.29 |
| Poultry manure | 26.70 | 185.91 | 314.19 | 65.55 | 19.77 | 0.39 a | 0.20 |
| Insect frass | 25.18 | 196.72 | 297.24 | 55.85 | 17.98 | 0.36 a | 0.27 |
| Vinasse-based fertilizer | 25.14 | 128.06 | 254.83 | 51.59 | 14.52 | 0.25 b | 0.18 |
| MSE | 16.44 | 9511.57 | 4866.77 | 177.74 | 17.32 | 0.002 | 0.011 |
| CV (%) | 16.55 | 52.32 | 23.02 | 23.48 | 28.81 | 19.31 | 44.78 |
| Tukey (5%) | 8.51 | 204.81 | 146.50 | 28.00 | 8.74 | 0.08 | 0.22 |

MSE = mean square error; CV = coefficient of variation; Tukey (5%) = Tukey's least significant difference (5%). Different letters within each column indicate significant differences according to Tukey's range test (5%).

The effect of fertilizer treatments on non-essential trace elements in lettuce leaves was significant for the heavy metals Co and Pb while there were no significant differences for Na, Al, Ba, Cd, Cr, Se and Sn (Table 6). The vinasse-based fertilizer was able to reduce significantly ($p < 0.01$) the amount of Co in lettuce leaves compared to the control which had the highest value. Moreover, the soil amendment with the three organic fertilizers leads to a significant ($p < 0.01$) reduction in leaf Pb concentration in comparison to unfertilized control.

**Table 6.** Effect of fertilizer treatments on non-essential trace elements in lettuce leaves.

| Fertilizer Treatments | Non-Essential Trace Elements (mg/kg DW) | | | | | | | | |
|---|---|---|---|---|---|---|---|---|---|
| | **Na** | **Al** | **Ba** | **Cd** | **Co** | **Cr** | **Pb** | **Se** | **Sn** |
| Control | 3.62 | 440.17 | 19.04 | 0.24 | 0.23 a | 0.28 | 1.29 a | 0.92 | 0.89 |
| Poultry manure | 3.83 | 402.93 | 22.28 | 0.36 | 0.17 ab | 0.32 | 0.81 b | 0.84 | 0.83 |
| Insect frass | 4.66 | 338.47 | 25.35 | 0.26 | 0.17 ab | 0.40 | 0.77 b | 0.98 | 0.83 |
| Vinasse-based fertilizer | 3.61 | 254.33 | 17.63 | 0.28 | 0.09 b | 0.25 | 0.52 b | 1.02 | 0.83 |
| MSE | 1.127 | 32,475.66 | 42.56 | 0.004 | 0.001 | 0.027 | 0.044 | 0.034 | 0.004 |
| CV (%) | 26.68 | 49.27 | 31.32 | 25.99 | 37.12 | 50.66 | 40.59 | 19.04 | 7.48 |
| Tukey (5%) | 2.23 | 378.44 | 13.70 | 0.13 | 0.08 | 0.35 | 0.44 | 0.39 | 0.14 |

MSE = mean square error; CV = coefficient of variation; Tukey (5%) = Tukey's least significant difference (5%). Different letters within each column indicate significant differences according to Tukey's range test (5%).

Soil active carbon at the end of the trial was significantly ($p < 0.001$) affected by fertilizer treatments ($p < 0.001$) with higher values with poultry manure (444.3 mg C/kg) and insect's frass (485.8 mg C/kg) in comparison with vinasse-based fertilizer (375.7 mg C/kg) and unfertilized control (369.5 mg C/kg).

In the current experiment, the application of poultry manure and insect's frass significantly increased the AcP ($p < 0.001$), AlkP ($p < 0.001$) and DHA ($p < 0.001$) in comparison to the control and vinasse-based fertilizer, which were not significantly different (Table 7). The ArS was significantly ($p < 0.01$) highest after the application of insect's frass, followed by poultry manure and vinasse-based fertilizer which had similar values while the lowest value was observed in the unfertilized control. The NAGase—Chitinase had a significantly ($p < 0.001$) higher value in insect's frass treatment in comparison to the control and vinasse-based fertilizer treatment while no significant difference was recorded with poultry manure application. In case of applying vinasse-based fertilizer, NAGase—Chitinase had significantly higher value than unfertilized control. THA had significant ($p < 0.001$) higher values in insect's frass and poultry manure than in control while vinasse-based fertilizer had an intermediate value (Table 7). A significant Pearson correlation was observed between organic fertilizer rate and soil enzyme activity for AcP (r = 0.981 *), ArS (r = 0.938 *), DHA (r = 0.933 *), and THA (r = 0.955 *). Moreover, no significant correlation between organic fertilizer rate and soil enzyme activity was observed for AlkP and NAGase—chitinase.

Hierarchical clustering analysis coupled with a heatmap was performed to provide a visual representation of the changes in morpho-physiological crop traits, soil active carbon, and soil enzymatic activity after the application of the three organic fertilizer treatments (Figure 3). In the treatment dendrogram the unfertilized control was clearly separated from the other fertilizer treatments due to the decrease of shoot and root growth, leaf SPAD index at 9 and 15 days after transplanting (DAT), leaf concentration of P, K, Mg, B, Cu, Cd, soil enzymatic activity (AcP, AlkP, ArS, DHA, NAGase, THA), and soil active C, and the increase of leaf concentration of N, Pb, Ca, Co, Sn and root to shoot ratio. Moreover, vinasse-based fertilizer treatment was clearly separated from the two other organic fertilizers due to the increase of root dry weight, leaf SPAD index at 9, 15, 23, and 28 DAT and leaf concentration of P, and the decrease of leaf concentration of Ca, Mg, Fe, Mn, Zn, Cu, Mo, Al, Ba, Cr, Na, Pb, Co, Ni, soil enzymatic activity (AcP, AlkP, ArS, DHA, NAGase, THA) and soil active C. Insect frass and poultry manure were clustered together on the same branch; the main differences between these two organic fertilizers were due to the increase of shoot fresh and dry weight, maximum quantum yield of PSII (Fv/Fm) of leaves at 20, 25, and 30 DAT,

leaf SPAD index at 4 DAT, and leaf concentration of Se, Cr, Ba, Na, Ni, and the decrease of root to shoot ratio, leaf concentration of N, NO$_3$, K, Zn, and Cd in insect frass compared to poultry manure treatment.

**Table 7.** Effect of fertilizer treatment on soil enzyme activity per gram of dry soil per hour in greenhouse lettuce trial.

| Fertilizer Treatment | AcP (µg p-Nitrophenol/g h) | AlkP (µg p-Nitrophenol/g h) | ArS (µg p-Nitrophenol/g h) | DHA (µg Triphenyl formazan/g h) | NAGase—Chitinase (µg p-Nitrophenol/g h) | THA (µg Hydrolyzed FDA/g h) |
|---|---|---|---|---|---|---|
| Control | 48.74 b | 362.25 b | 4.83 c | 0.42 b | 8.84 c | 14.26 c |
| Poultry manure | 91.61 a | 619.94 a | 8.29 b | 1.87 a | 47.81 ab | 63.93 a |
| Insect frass | 99.35 a | 561.58 a | 11.43 a | 1.88 a | 57. 61 a | 61.31 a |
| Vinasse-based fertilizer | 65.35 ab | 336.50 b | 8.21 b | 0.84 b | 30.85 b | 35.56 b |
| MSE | 298.53 | 1766.50 | 1.81 | 0.19 | 71.26 | 26.31 |
| CV (%) | 34.13 | 28.14 | 32.90 | 61.52 | 56.66 | 49.10 |
| Tukey (5%) | 36.28 | 88.26 | 2.83 | 0.93 | 17.73 | 10.77 |

AcP = acid phosphatase; AlkP = alkaline phosphatase, ArS=arylsulfatase, N-acetyl-β-D-glucosaminidase (NAGase-chitinase); DHA = dehydrogenase activity; THA = total hydrolase activity. MSE = mean square error; CV = coefficient of variation; Tukey (5%) = Tukey's least significant difference (5%). Different letters within each column indicate significant differences according to Tukey's range test (5%).

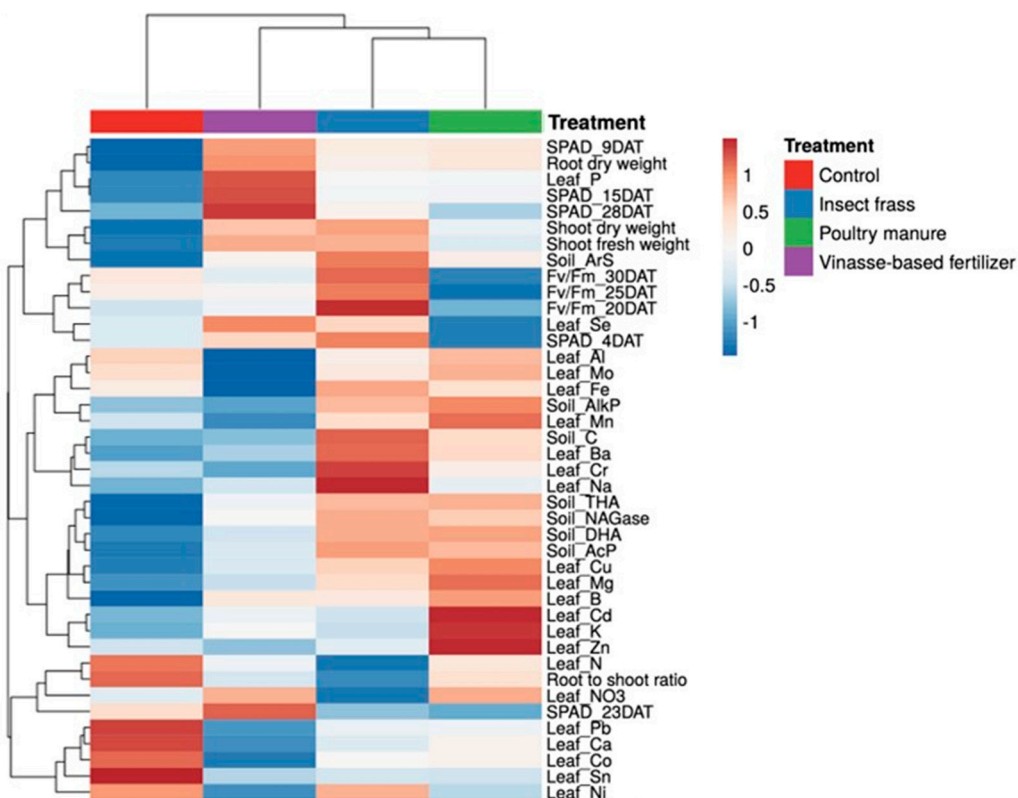

**Figure 3.** Cluster heat map analysis summarizing lettuce plants response to organic fertilizer application. Original values are ln (x + 1)-transformed. Columns and rows are clustered using Euclidean distance and complete linkage.

## 4. Discussion

The rate and timing of mineralization of organic fertilizer is a key factor to meet the plant's needs for nutrients. Mineralization process of organic fertilizers is affected by several factors, such as quality of organic matter, soil temperature, soil water availability, etc. Among these factors, C/N ratio of organic fertilizer plays a pivotal role in mineralization and immobilization processes of N in soils. The lower the C/N ratio, the faster nitrogen is supplied into the soil for direct crop absorption [42]. Our data showed in line with several authors [43–45], that vinasse-based fertilizer had the highest N mineralization rate. These results can be linked to its lower C/N ratio (4.8) in comparison to poultry manure (10.2) and insect frass (14.3). Therefore, vinasse-based fertilizer is a more suitable fertilizer than poultry manure and insect' frass in terms of supplying the mineral N in soil during the short crop cycle of lettuce under greenhouse conditions.

Leaf chlorophyll concentration has been linked to plant stressand nutrient status. As a component of chlorophyll molecule, nitrogen is a major nutrient affecting leaf chlorophyll concentration. The SPAD meter was demonstrated to be a useful tool for non-destructively assessing leaf chlorophyll concentration. In the greenhouse trial, leaf SPAD index at 4 DAT was not significantly affected by treatments, indicating that a significant proportion of absorbed nitrogen during the early stages of plant growth may have been used to generate structural elements of the plant rather than chlorophyll as suggested by several authors [46]. Moreover, the significant difference in chlorophyll concentration at 9 and 15 DAT could be related to the faster N mineralization of vinasse-based fertilizer in comparison to other treatments and to the high plant demand of nitrogen during the exponential phase of plant growth [47]. On the other hand, the non-significance among all treatments at 23 and 28 DAT could be explained by the low N demand by the lettuce plants during this final growth stage.

Maximum quantum yield of PSII (Fv/Fm) is a sensitive indicator of plant photosynthetic ability of leaves where low Fv/Fm values may indicate downregulation of photosynthesis resulting from stress and/or photoinhibition [48,49]. In this current study, the Fv/Fm values observed during the growing cycle in all treatments were in the range of 0.75 to 0.85 reported for healthy leaf crops [50,51].

According to the FAO data of crop salt tolerance, lettuce is considered a moderately sensitive crop to salinity with a salinity threshold value for shoot fresh weight of 1.3 dS/m and a percentage of shoot yield reduction per dS/m above the threshold value of 13. Considering that the soil EC of saturated paste extract was 5.47 dS/m, the relative crop yield reduction determined using the Maas–Hoffman model [52] was 45.8% indicating a strong negative impact of soil salinity on shoot fresh biomass in the experimental plots. Soil amendment with organic fertilizers increased shoot fresh weight of lettuce especially in vinasse-based and insect's frass treatments. Shoot dry weight showed a similar behavior of shoot fresh weight, indicating that the increase of shoot fresh weight under organic fertilization regimes in comparison with unfertilized control was due to an enhancement of carbon assimilation. This finding indicates an important role of the three organic fertilizers in mitigating the negative impact of salinity on lettuce growth in comparison to the unfertilized control.

Organic matter can be incorporated into soil to provide a positive impact on the physical, chemical, and biological characteristics of the soil [53–56]. These benefits are mainly ascribed to the supply of organic compounds and especially humic substances through the organic amendment. Several authors reported that humic substances can increase aggregate stability in soil [57], stimulate secondary metabolism pathways involved in stress response [58], immobilize heavy metals by binding or redox reaction [59–61], stimulate the plant growth, biosynthetic activity, and nutrient uptake [62,63] activate antioxidative enzymatic function, and control the reactive oxygen species content in plants [64]. Moreover, organic fertilization can promote beneficial microflora as the result of the microbial cells' ability to adapt to osmotic stress by creating osmolytes, which counteract osmotic stress, thanks to the availability of carbon from the additional organic matter [53,65,66].

Nitrate concentration of lettuce leaves is an important quality trait affecting marketability of product. The Commission Regulation EU No 1258/2011 [67] amending Regulation EC No 1881/2006 established maximum levels for nitrates in lettuce grown in greenhouse from 1st October until 31st March to be 5000 mg kg$^{-1}$ of fresh weight. In the current study, nitrate concentrations in lettuce leaves were always lower than the EU threshold value indicating no restriction for commercialization. However, insect's frass treatment gave a lower leaf nitrate concentration in comparison with the other treatments, indicating a positive role of insect's frass in the activation of key enzymes for nitrogen assimilation in plant cells (e.g., nitrate reductase).

Organic matter is the major component of organic fertilizer which increases the soil's macronutrient and micronutrient concentrations, and through enhancing its structure, it leads to increased soil fertility and quality [68]. Considering the effect of organic fertilizers on macro and micronutrients concentration in lettuce leaves, significant differences were recorded only on leaf concentration of P and Mo, which were the highest and the lowest in vinasse-based treatment, respectively. The increase of leaf P concentration in vinasse-based treatment can be related to the different P supplied to the plants which was higher with vinasse-based fertilizer (52.4 kg P ha$^{-1}$) than poultry manure and insect's frass (both 39.3 kg P ha$^{-1}$) and untreated control. Moreover, the fast mineralization of vinasse-based fertilizer may also have enhanced the soil P availability for plant uptake. According to the sufficiency range for butterhead lettuce under greenhouse conditions [69], the macronutrients and micronutrients concentrations in lettuce leaves were in the sufficient ranges as for N, Mg, Fe, Zn, and Mo, while they exceeded the sufficient range as for K, Ca, Mn, and Cu. Leaf P concentrations in all treatments were below the sufficient range reported by the authors [69], indicating the P availability was probably a critical factor in plant growth under the current experiment.

Regarding the effect of organic fertilizers on non-essential trace elements in lettuce leaves, the reduction of leaf Co concentration by vinasse-based fertilizer application and the decrease of leaf Pb concentration by the soil amendment with the three organic fertilizers indicate a positive role of organic matter in mitigating the accumulation of some heavy metals in plant tissues due to adsorption and chelating actions of heavy metals ions [70–72]. Heavy metals are immobilized by humic substances primarily by binding (complexation, ion exchange, and physical adsorption) or redox reaction [59–61]. Binding heavy metals can fix and limit their fluidity in the soil, whereas heavy metal redox reactions can diminish or remove their toxicity [73]. Moreover, plant uptake of heavy metals can be restricted by the increasing of microorganisms in organically fertilized soil [70,71,74,75]. Similar results were reported in an incubation experiment where amendment with organic fertilizers was effective in reducing the availability of polluted soil with the heavy metal Cd, and the inhibitory effect was stronger as the quantity of organic fertilizer was augmented [71].

Soil enzymes are mostly microbial in origin and are tightly related to microbial abundance and activity. They are critical in catalyzing reactions required for organic matter decomposition and nutrient cycling [76]. Climate conditions, type of amendment, cultivation methods, crop type, and edaphic qualities are all factors that influence enzyme catalyzed reactions. Soil enzyme activities are mainly influenced by organic matter content [77] and are frequently used as indicators of microbial activity and soil fertility [78]. In this study, soil enzymes responsible for the major nutrient cycles (N, P, C and S) were chosen. Phosphatases (AcP, AlkP) are very useful enzymes in agriculture because they hydrolyze organic phosphorus compounds and convert them into inorganic phosphate that plants can absorb [79]. Arylsulfatase (ArS) is an enzyme that participates in the hydrolysis of arylsulphate esters by the fission of the oxygen-sulfur (O-S) bond. This enzyme is known to play a role in the mineralization of ester sulfate in soils [80]. Furthermore, because only fungi (not bacteria) possess ester sulfate, the substrate of arylsulfatase [81,82], it may be an indirect indication of fungal abundance in the soil. Dehydrogenases (DHA) are among the most significant soil enzymes for determining biological activity in the soil because they represent the entire oxidative metabolic activity of soil microbial community [83]. N-acetyl-β-D-glucosaminidases (NAGase—Chitinase) are abundant in nature, and the low molecular

weight amino sugars produced by their hydrolysis constitute an essential source of nitorgen for soil microbes. β-glucosidase catalyzes the hydrolysis of β-D-glucopyranosides and is one of three or more enzymes necessary for the saccharification of cellulose [84,85]. The total hydrolytic activity (THA) measurement is a simple method for assessing total microbial activity [41]. Several authors have reported an increase in soil microbial biomass, bacteria populations, fungi and actinomycetes, as well as soil enzyme activities after applying organic fertilizers such as manures and composts [6,8,27,28,86]. In agreement with the above findings, soil enzyme activities increased in organically treated plots especially with the application of poultry manure and insect's frass. The soil enzymatic activity for AcP, ArS, DHA, and THA was positively correlated with the organic fertilizer rate while no significant correlation between organic fertilizer rate and soil enzymatic activity was observed for AlkP and NAGase—chitinase, indicating that the quality of organic matter supplied by fertilizers may have differentially affected their activity. Several authors found that the quality of organic matter may alter the organization of the soil microbial population because microorganisms have substrate preferences [87]; the different structure of microbial community resulting from organic fertilizer application can explain the variation of the soil enzyme activity and the lack of significant correlation for AlkP and NAGase.

The available or labile portion of soil organic matter, also known as active carbon, is a sensitive tool for measuring soil carbon change. Active carbon is used as a readily available food and energy source for the soil microorganisms. At the end of the trial, organic fertilization with poultry manure or insect's frass was able to raise the soil active carbon in comparison to vinasse-based fertilizer and unfertilized control. It has been demonstrated that soil amendment with high C/N ratio may improve and maintain soil labile C for a longer period [88]. Therefore, the highest C/N ratio of poultry manure and insect's frass together with the highest fertilizer rate can explain the increase of soil active carbon at the end of the trial compared to vinasse-based fertilizer.

## 5. Conclusions

Sustainable intensification of greenhouse vegetable production relies on the application of organic matter to maintain soil fertility and support plant nutrition providing that nutrient release in available forms for plant uptake match the crop nutrient requirements. Based on the laboratory incubation assay, the vinasse-based fertilizer can be recommended for supporting plant nutrition due to the fast mineralization and release of mineral N which is highly required during the first growing period of the short lettuce cultivation cycle. Our data demonstrated that application of organic fertilizers such as poultry manure, insect's frass, and vinasse-based fertilizer in greenhouse lettuce production was a good strategy for reducing stressful conditions arising from the soil salinization and soil organic matter depletion due to intensive vegetable crop production. Organic fertilizers enhanced lettuce growth as demonstrated by the increase of shoot fresh and dry weight. Moreover, insect frass was also effective in improving the lettuce quality by reducing the nitrate concentration in leaves. In terms of soil quality, poultry manure and insect's frass were more effective in restoring soil biological fertility as demonstrated by the increase of soil active carbon and the enzyme activity involved in nutrient cycling.

Finally, further research is needed to investigate the potential benefits of combined applications of vinasse-based fertilizer and the other organic fertilizers for supporting plant nutrition and at the same time enhancing the beneficial effects on soil fertility in lettuce production.

**Author Contributions:** Conceptualization, M.C. and G.C.; methodology, M.C., P.I., P.B. and G.C.; validation, M.C., A.E.C., P.I., P.B. and G.C.; formal analysis, M.C., P.I., Y.R. and G.C.; investigation, M.C., P.I., P.B. and G.C.; resources, M.C., P.I. and G.C.; data curation, M.C., A.E.C., P.I., Y.R. and G.C.; writing—original draft preparation, M.C., A.E.C. and G.C.; writing—review and editing, M.C., A.E.C., Y.R., P.B. and G.C.; visualization, M.C., A.E.C. and G.C.; supervision, M.C. and G.C.; project administration, M.C. and G.C.; funding acquisition, G.C. All authors have read and agreed to the published version of the manuscript.

**Funding:** This work was partially funded by MIUR in the frame of the initiative "Departments of excellence", Law 232/2016.

**Data Availability Statement:** Data available on request from the corresponding author.

**Acknowledgments:** We thank Antonio Fiorillo and Rares Balan for their assistance in supporting greenhouse and laboratory assays. We also thank Hello Nature for providing the tested organic fertilizers.

**Conflicts of Interest:** The authors declare no conflict of interest.

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
