# Peer review of "Organic Fertilizer Sources Distinctively Modulate Productivity, Quality, Mineral Composition, and Soil Enzyme Activity of Greenhouse Lettuce Grown in Degraded Soil"

_agronomy, doi:10.3390/agronomy13010194_

Round 1

Reviewer 1 Report

Dear Sir/Madam

Thank you so much for your effort

Article Organic fertilizer sources distinctively modulate productivity, 

quality, mineral composition, and soil enzyme activity of greenhouse lettuce grown in degraded soil was studied. Very well done. The article is also well organized. But to improve the article, consider the following:

lines 112 to 114 should be deleted (We hypothesized that the application of fast-acting organic fertilizers could have a positive impact on soil fertility indicators like enzymatic activities, and quali-quantitative traits of lettuce under low fertile soil resulting from intensive greenhouse vegetable production duction. T)

There is a problem with this article. You somehow showed the analysis of variance and comparison of means in a table, and a lot of information in the article has been removed, and the reader can't get proper information with the numbers in the tables. The authors show the two issues of analysis of variance and comparison of averages separately. In addition, in the analysis of variance table, the means square and coefficient of variation must be specified so that the accuracy of the tests can be determined.

In order to show the relationship between traits in a good way, the correlation between traits should be shown in the form of a heat map.

Author Response

Dear Reviewer

Thank for considering the manuscript well done and well organized and for your comments.

As suggested, Lines 112-114 have been deleted in the revised version of the manuscript.

Moreover, tables and figures have been improved inserting the mean square errors, coefficients of variation and Tukey’s least significant differences (5%) whereas the P values from the ANOVA results have been inserted in the text of the revised version of the manuscript.

In agreement with your comments, the heat map has been inserted as figure 2 in the revised version of the manuscript.

Minor corrections hve been also incuded during the revision process.

Best regards

Giuseppe Colla

Reviewer 2 Report

A good study done on Organic fertilizer sources distinctively modulate productivity, 2 quality, mineral composition, and soil enzyme activity of greenhouse lettuce grown in degraded soil. The study is intresting study which reflect organic fertilizers impact. Data presentation work is nicely done. well written manuscript. 

However, some suggestion that can be consider 

can it possible to give total carbon or total organic carbon content data of organic fertilizer sources. 

Conclusions can be sort with significant findings  

Author Response

Dear Reviewer

Thank for considering the study interesting and the data well-presented and the manuscript well-written.

As suggested, total carbon supplied with the different fertilizer sources was provided in the M&M section (Lines  183-185). The total carbon values were calculated considering that poultry manure had 40 g N in 1 kg of fertilizer and there were 10.2 g of carbon for each 1 g of nitrogen in the fertilizer, insect frass had 30 g N in 1 kg of fertilizer and there were 14.3 g of carbon for each 1 g of nitrogen in the fertilizer, and vinasse-based fertilizer had 60 g N in 1 kg of fertilizer and there were 4.8 g of carbon for each 1 g of nitrogen in the fertilizer.

In agreement with your comments, the conclusions have been shortened leaving the main results of the work.

Minor corrections have been also inserted  during the revision process.

Best regards

Giuseppe Colla

Reviewer 3 Report

This manuscript has investigated the efect of organic fertilizer application on soil properties and lettuce growth in degraded soil, which has some guidance meaning for soil fertility recover of degraded soil in greenhouse conditions. But, the whole manuscript quality of current Version does not meet the requirements of the journal. Some detail suggestions were listed in order to impove the manuscript.

1. Over all, the article is more like a research report than a study paper. This article is absent in logic, and some language expressions are inaccurate and unscientific. For example, Line 105, line 139-140, Line 180-181, and so on.

2. Example design and sample analysis should be seperated for 2.1 and 2.2 sections. 

3. The description of the experimental design is unclear. There is no sampling time for 2.1 section. The order of descriptions is confusing, resulting in readers can not be able to obtain useful information for 2.2.

4. Result section, the titles are not accurate, it will be better if authors can use the highlights as titiles here. 

5. The presentation of results is too fragmented,which is not suitable for rigor and scientific rigor of research papers.

Author Response

Dear Reviewer

Thank for your comments; below you can find a detailed response to each comment.

  1. Over all, the article is more like a research report than a study paper. This article is absent in logic, and some language expressions are inaccurate and unscientific. For example, Line 105, line 139-140, Line 180-181, and so on.

Dear Reviewer

Let me draw your attention that the manuscript has been prepared following the guide for authors indicated by Agronomy journal. Moreover, we believe that the manuscript follows a logical flow with in-depth discussion of the results in the light of the existing literature. Concerning lines 105, lines 139-140, and lines 180-181, we tried to improve the language expressions in the revised version of the manuscript.

  1. Example design and sample analysis should be separated for 2.1 and 2.2 sections.

Dear Reviewer following your suggestions we separated the experimental design, measurements and analysis in the two sections (2.1 and 2.2) of the revised manuscript.

  1. The description of the experimental design is unclear. There is no sampling time for 2.1 section. The order of descriptions is confusing, resulting in readers can not be able to obtain useful information for 2.2.

Dear Reviewer, we improved the description of the experimental design indicating the sampling times in sub-section 2.1.2. Moreover, we improved the description of the trials in the M&M section.

  1. Result section, the titles are not accurate, it will be better if authors can use the highlights as titiles here.

Dear Reviewer, following your suggestion the titles of result sections have been changed to better introduce the results reported in each section.

  1. The presentation of results is too fragmentedwhich is not suitable for rigor and scientific rigor of research papers.

Dear Reviewer, we used the following logical approach in presenting the results:

  • in section 3.1, we presented the different behavior of organic fertilizers in term of N mineralization under laboratory incubation conditions;
  • in section 3.2 we reported the results of greenhouse agronomic trial starting from the effects of treatments on physiological parameters of lettuce crop during the growing cycle (SPAD index, Fv/Fm), then we displayed the final crop biomass, leaf quality and leaf mineral concentration at harvest, and finally we reported the results of the soil enzymatic activity which was measured at the end of the trial.

We believe that the above presentation flow of the results can help the reader to better understand the role of tested organic fertilizers in supporting plant nutrition (N mineralization rate) and improving soil fertility (soil enzymatic activity) leading to an enhancement of quali-quantitative traits of lettuce crop under degraded soil conditions.

Other minor corrections have been inserted in the revised manuscript.

Best regards

Giuseppe Colla

Reviewer 4 Report

Manuscript Number: agronomy-2118763-peer-review-v1

Title: Organic fertilizer sources distinctively modulate productivity, quality, mineral composition, and soil enzyme activity of greenhouse lettuce grown in degraded soil.

Comments

This present study was carried out to assess the potential role of organic fertilizer sources distinctively modulate productivity, quality, mineral composition, and soil enzyme activity of greenhouse lettuce grown in degraded soil. However, the main drawbacks of this article are lack of the novelty, no deep mechanism discussed and poor write-up. Therefore, this article is not suitable to be published in a high quality journal i.e. Agronomy MDPI. Before submit to any other Journal some specific comments are given below.

1.      Introduction is not well written please try to revise it with latest published articles.

2.      Novelty point of view, authors have suggested to mention, how this present study is unique from already published work.

3.      Authors have performed a short-term study and the present data is not enough and no any correlation e.g. RDA was mentioned in the studied parameters.

4.      Discussion section need to be strengthen with relevant and latest literature.

5.      The figures expression is not good please try to use colored figures and improve the quality significantly.

6.      Finally, the language of the manuscript should be improved to increase the readability of the manuscript.

7.      Overall, I also see that quality of the paper is not suitable for publication in a top quality journal like: Agronomy MDPI. Hence, it is rejected.

Author Response

Dear Reviewer

Below you can find answers to your comments.

  1. Introduction is not well written please try to revise it with latest published articles.

Dear Reviewer, we believe that the 'Introduction' follows a logical flow starting from the introduction of the problem to be addressed, possible solutions from scientific literature and advances of scientific knowledge generated by our work. Moreover, the cited literature includes several articles published in the last years (e.g., n. 20 in 2022; n. 21 in 2020).

  1. Novelty point of view, authors have suggested to mention, how this present study is unique from already published work.

Let me draw you attention, Dear Reviewer, that several articles have been published on organic fertilization of lettuce. However, many of these papers are based on research trials conducted in pots without considering the potential benefits of organic fertilizers in degraded soil resulting from intensive greenhouse vegetable crop production (soil salinization and loss of organic matter). Moreover, limited studies have been conducted considering advanced sustainable organic fertilizers such as insect frass which is becoming popular worldwide due to the increase of insect farming for human and animal nutrition. Finally, the innovation of our study is also related to the in-depth study of many morpho-physiological crop traits, soil active carbon and soil enzymatic activity in order to better understand the impact of organic fertilizers on the crop-soil system.   

  1. Authors have performed a short-term study and the present data is not enough and no any correlation e.g. RDA was mentioned in the studied parameters.

Dear Reviewer, we believe that the experimental data, which includes two experiments (one laboratory experiment and one greenhouse trial), are enough for justifying the publication in Agronomy. Thirty-five parameters have been measured in the greenhouse trial for providing a comprehensive view of the effects of the organic fertilizers on the crop-soil system. Moreover, in the revised version of the manuscript, a hierarchical clustering analysis coupled with a heatmap was performed to provide a visual representation of the changes in morpho-physiological crop traits, soil active carbon and soil enzymatic activity after the application of the three organic fertilizer treatments.

  1. Discussion section need to be strengthened with relevant and latest literature.

Dear Reviewer, we believe that the most relevant literature on this topic has been cited. However, if you have any additional suggestion on relevant literature to be inserted in the discussion section, please let us know.

  1. The figures expression is not good please try to use colored figures and improve the quality significantly.

Thank for this comment. We inserted the colors in the figures to improve their quality.

  1. Finally, the language of the manuscript should be improved to increase the readability of the manuscript.

We improved the English language and we checked it with a native-speaker

  1. Overall, I also see that quality of the paper is not suitable for publication in a top quality journal like: Agronomy MDPI. Hence, it is rejected

Based on our previous comments, we believe that the quality of the manuscript meets the standards of Agronomy.

Best regards

Giuseppe Colla

Round 2

Reviewer 1 Report

Thanks

Reviewer 3 Report

Authors have carefully revised the manuscript, current version is good enough for publishing.

Reviewer 4 Report

Manuscript Number: agronomy-2118763-peer-review-v1

Title: Organic fertilizer sources distinctively modulate productivity, quality, mineral composition, and soil enzyme activity of greenhouse lettuce grown in degraded soil.

Comments

Thank you so much, I have carefully checked your revised manuscript. Authors have paid more attention and carefully revised manuscript. Here, I am accepting revised manuscript and recommended it for publication to Agronomy MDPI Journal.